# Plant-Derived Pesticides as an Alternative to Pest Management and Sustainable Agricultural Production: Prospects, Applications and Challenges

**DOI:** 10.3390/molecules26164835

**Published:** 2021-08-10

**Authors:** Augusto Lopes Souto, Muriel Sylvestre, Elisabeth Dantas Tölke, Josean Fechine Tavares, José Maria Barbosa-Filho, Gerardo Cebrián-Torrejón

**Affiliations:** 1Programa de Pós-Graduação em Produtos Naturais e Sintéticos Bioativos, Universidade Federal da Paraíba, João Pessoa 58051-900, Brazil; augustosouto@gmail.com (A.L.S.); josean@ltf.ufpb.br (J.F.T.); jbarbosa@ltf.ufpb.br (J.M.B.-F.); 2COVACHIM-M2E Laboratory EA 3592, Department of Chemistry, Fouillole Campus, University of the French West Indies, UFR Sciences Exactes et Naturelles, CEDEX, 97157 Pointe-à-Pitre, France; muriel.sylvestre@univ-antilles.fr; 3Departamento de Botânica, Instituto de Biociências, Universidade de São Paulo, São Paulo 05508-090, Brazil; elisabeth.tolke@gmail.com

**Keywords:** biopesticides, bio-based pesticides, chemical ecology, pest control, natural products

## Abstract

Pests and diseases are responsible for most of the losses related to agricultural crops, either in the field or in storage. Moreover, due to indiscriminate use of synthetic pesticides over the years, several issues have come along, such as pest resistance and contamination of important planet sources, such as water, air and soil. Therefore, in order to improve efficiency of crop production and reduce food crisis in a sustainable manner, while preserving consumer’s health, plant-derived pesticides may be a green alternative to synthetic ones. They are cheap, biodegradable, ecofriendly and act by several mechanisms of action in a more specific way, suggesting that they are less of a hazard to humans and the environment. Natural plant products with bioactivity toward insects include several classes of molecules, for example: terpenes, flavonoids, alkaloids, polyphenols, cyanogenic glucosides, quinones, amides, aldehydes, thiophenes, amino acids, saccharides and polyketides (which is not an exhaustive list of insecticidal substances). In general, those compounds have important ecological activities in nature, such as: antifeedant, attractant, nematicide, fungicide, repellent, insecticide, insect growth regulator and allelopathic agents, acting as a promising source for novel pest control agents or biopesticides. However, several factors appear to limit their commercialization. In this critical review, a compilation of plant-derived metabolites, along with their corresponding toxicology and mechanisms of action, will be approached, as well as the different strategies developed in order to meet the required commercial standards through more efficient methods.

## 1. Introduction

Pesticides may be defined as any compound or mixture of components intended for preventing, destroying, repelling or mitigating any pest [1]. Additionally, herbicides or weed-killers may also be considered as pesticides, and are used to kill unwanted plants in order to leave the desired crop relatively unharmed and well provided with nutrients, leading to a more profitable harvest [2].

Nevertheless, the world food production is constantly affected by insects and pests during crop growth, harvest and storage. As a matter of fact, there is an estimated loss of 18–20% regarding the annual crop production worldwide, reaching a value of more than USD 470 billion [3]. Furthermore, insects and pests not only represent a menace to our homes, gardens and reservoirs of water, but also, they transmit a number of diseases by acting as hosts to some disease-causing parasites. Therefore, the mitigation or control of pests’ activities may lead to a substantial reduction of the world food crisis as well as the improvement of human and animal health [2].

The great demand for food has led to the intensification of agricultural technology in order to achieve maximum productivity per hectare, through expansion of irrigation facilities, introduction of high-yielding varieties and application of increased amounts of agrochemicals, mostly synthetic [1] (for example, the 1,1,1-trichloro-2,2-bis(4-chlorophenyl)-ethane (DDT) used from 1939 to 1962 [4] and the polychlorinatedbiphenyls (PCBs) used from 1926 to 1970 [5]). In spite of various technological achievements over the years, serious problems have come along, especially due to the indiscriminate use of synthetic pesticides. As they remain in our planet for an extremely long period, its long persistence in our biosphere allows insects to develop resistance against them; they are also known to contaminate indispensable resources, such as water, air and soil. This is illustrated by the problem of chlordecone (CLD), a pesticide widely used from 1972 to 1993 in the French West Indies (FWI) [6] which is often called as the “monster of the Antilles” because of the extent of soil and biomass contamination. Chemically, CLD is an organochlorine ketone, with high steric hindrance and high hydrophobicity, which allows it to adsorb strongly into soils rich in organic matter. At the same time, it is non-volatile and has a low biodegradability. It has also been demonstrated that CLD is a particularly persistent molecule, which remains almost perennial in soils, and the phenomenon of bioaccumulation in living organisms has been observed. Despite the ban on its use in the early 1990s, this molecule is still present today in the waters and soils of the French West Indies [7]. Extremely few pesticides used in the post-DDT era have long half-lives in the environment and none bioaccumulate in the way that the organochlorines did. Moreover, most of the synthetic pesticides act against non-target organisms (mammals, fish and plant species), becoming a potential health hazard to consumers. Furthermore, they are too expensive for the farmers in developing countries [8,9,10,11].

This problematic is illustrated as well by the neonicotinoid insecticides (such as imidacloprid, clothianidin, thiamethoxam, acetamiprid and thiacloprid). This more recent class of molecules is active against several pests (targeting the acetylcholine receptors) and can be applied to different cultures (such as tobacco, cotton, peach and tomato) [12].

The use of neonicotinoid insecticides was allowed in Europe in 2005 until 2013 (see Regulation (EU) No 485/2013) when the employ was restricted to protect wildlife, such as pollinators (honeybees), mammals, birds, fish, amphibians and reptiles, and the effects on vertebrates—mammals, birds, fish, amphibians and reptiles [13].

This awareness regarding pest problems and the environment has led to the search for powerful and eco-friendly pesticides that degrade after some time, avoiding pest resistance, which is also pest-specific, non-phytotoxic, nontoxic to mammals and relatively less expensive in order to obtain a sustainable crop production [14,15]. In addition to the awareness achieved in the various countries (such as China, United States, Brazil or Turkey) [16,17] and also in European Union (EU), the legislation has become increasingly stringent and binding with the consequences of a green choice in the use of pesticides and approbation of plant derivatives allowed in the biological control regulation (for example: clove (*Syzygium aromaticum* [L.] Merr. and L.M. Perry [Myrtaceae]) essential oil in the EU, the derived terpenes from *Chenopodium ambrosioides* L. (Amaranthaceae) in US, Ginkgo (*Ginko biloba* L. [Ginkgoaceae]) fruit extract and *Psoralea corylifolia* L. (Leguminosae) seed extract in China or the extract of *Tephrosia candida* DC. (Fabaceae) in Brazil).

In this context, biopesticides may meet those required standards and become the key to solve pest problems and promote sustainable production, once they are cheap, target-specific, less hazardous to human health, bio-degradable and therefore environmentally friendly. Biopesticides are pest management agents based on biochemicals derived from living microorganisms, insects and plants. [2,18,19]. In this review, we will focus on biopesticides from plant origin.

Among the plant-derived pesticides, there are the insecticides (constituents that kill insects in any stage of development: adults, ova and larvae), which act by several different mechanisms affecting one or more biological systems, including nervous, respiratory and endocrine systems, as well as water balance. Additionally, insecticides can also be classified depending on the mode of its entry into the insect, namely: stomach poisons, contact poisons and fumigants [11]. A compilation of natural insecticides with their corresponding toxicity and mechanisms of action may be found in Table 1.

## 2. Plant Derived Insecticides That Affect the Nervous System

The majority of insecticides, whether biological or synthetic, fit in this category, acting on several targets, such as: voltage-gated sodium channels, voltage-gated calcium channels, acetylcholinesterase enzyme (AChE), nicotinic acetylcholine receptors, GABA receptors and octopamine receptors.

### 2.1. Voltage-Gated Sodium Channels

Pyrethrum, an oleoresin extracted from the dried flowers of the pyrethrum daisy, *Tanacetum cinerariifolium* (Trevir.) Sch. Bip. (Asteraceae), contains two major compounds, namely Pyrethryns I and II (cyclopropylmonoterpene esters) [20] that act as a modulator on voltage-gated sodium channels, which are essential for proper electrical signaling in the nervous system, causing a delay in sodium channel closing, resulting in over neuroexcitation, leading to loss of control of the coordinated movement, paralysis and death. This mode of actions is highly similar to other synthetic insecticides (such as synthetic pyrethryns). However, natural pyrethrins used to be more target-specific than synthetic ones [21]. It is defined as a contact and stomach poison that provides an immediate knockdown when applied, which has also demonstrated low toxicity to mammals and a particularly short residual activity, once it is rapidly degraded by sunlight, air and moisture; therefore, frequent application may be required [22]. Pyrethrins may be used against a wide range of insects and mites, including spider mites, flies, fleas and beetles [23,24]. Its activity may be enhanced by incorporating piperonylbutoxide (PBO) as a synergist [2].

Other natural products with a similar mechanism of action have already been reported, such as decalesides I and II, firstly isolated from the roots of *Decalepis hamiltonii* Wight and Arn. (Apocynaceae). They are classified as trisaccharides, and are toxic to a variety of insects by contact exposure, not orally, but through contact to the gustatory receptors located in the tarsi of the insect [25].

Sabadilla, an insecticidal preparation from pulverized seeds of *Schoenocaulon officinale* (Schltdl. and Cham.) A. Gray ex Benth. (Melanthiaceae), has been used by native people from south and central America for many years. Its alkaloid preparation contains mainly two major alkaloids: cevadine and veratridine at a proportion of 2:1, which have a mode of action similar to pyrethrins, although the binding site seems to be different [26]. Sabadilla is one of the least toxic plant extracts with pesticide activity, considered a contact and stomach poison with minimal residual activity. However, its major alkaloids, when isolated, are much more toxic to humans, and affect mostly stinks, squash bugs, thrips, leafhoppers and caterpillars [11,27].

### 2.2. Voltage-Gated Calcium Channels

Ryanodine, an active component found in the roots and woody stems of *Ryania speciosa* Vahl (Salicaceae), native to Trinidad, activates the calcium channels from the sarcoplasmic reticulum of skeletal muscle cells. Once activated, the calcium channels release an excess of calcium ions into actin and myosin protein filaments, leading to skeletal muscle contraction and paralysis [28]. Ryanodine is a “fast act” poison, promoting its insecticidal activity either by contact or stomach, with a low mammalian toxicity and a long residual activity, providing up to two weeks of control after the first application. Ryania crude extracts insecticidal activity is synergized by piperonylbutoxide (PBO), and is reported to be most effective in hot waters, working efficiently against caterpillars, worms, potato beetles, lace bugs, aphids and squash bugs. [11,29]. Ryania is almost no longer used in the United States.

### 2.3. Acetylcholinesterase Enzyme (AChE)

Acetylcholinesterase (AChE), an enzyme that hydrolyzes the neurotransmitter acetylcholine, plays an important role regulating the transmission of the cholinergic nervous impulse, and may also be a target for biopesticides. Coumaran (2,3-dihydrobenzofuran), an active ingredient found in *Lantana camara* L. (Verbenaceae), inhibits this enzyme, building up the concentration levels of acetylcholine in the synapse cleft, causing an excessive neuroexcitation due to the prolonged biding of the neurotransmitter to its postsynaptic receptor, leading to restlessness, hyperexcitability, tremors, convulsion, paralysis and death [30]. It presents low toxicity to mammals and works rapidly against houseflies and grain storage pests, in spite of its short residual activity. Regarding its mechanism of action, coumaran may also be compared to the monoterpene 1,8-cineole [31] or other synthetic pesticides such as organophosphates and carbamates [32]. Moreover, Khorshid, et al. [33] have presented the inhibitory activity of methanolic extract from *Cassia fistula* L. (Fabaceae) roots and proposed the indole alkaloids as new potential active agents. However, it should be noted that many essential oils and terpenes therefrom have demonstrated anti-AChE activity in vitro, but their contribution to insect mortality is questionable [34].

### 2.4. Nicotinic Acetylcholine Receptors

In relation to nicotinic acetylcholine receptors, they are present in the insect nervous system, either on pre or postsynaptic nerve terminal, as well as the cell bodies of the inter neurons, motor neurons and sensory neurons [35]. Nicotine, an alkaloid firstly isolated from *Nicotiana tabacum* L. (Solanaceae), can mimic acetylcholine by acting as an agonist of the acetylcholine receptor, leading to an influx of sodium ion and generation of action potential. Under normal conditions, the synaptic action of acetylcholine is terminated by AChE. However, since nicotine cannot be hydrolyzed by AChE, the persistent activation caused by the nicotine leads to an overstimulation of the cholinergic transmission, resulting in convulsion, paralysis and finally death [35]. Nicotine is an extremely fast nerve toxin, most effective towards soft-bodied insects and mites. However, is the most toxic of all botanicals and extremely harmful to humans [2]. Alternatively, there is another class of insecticides inspired on nicotine chemical structure, called neonicotinoids, which may be represented by imidacloprid, acetamiprid and thiamethoxam. Similarly to nicotine, neonicotinoids interact with nicotinic acetylcholine receptors. However, they are more specific, being much more toxic to invertebrates such as insects than to mammals. Additionally, they have higher water solubility, which permits its application to soils and therefore, its absorption by plants, promoting a more efficient defense [36]. They may act as a contact or ingestion poison, leading to the cessation of feeding within several hours of contact followed by death shortly after [37].

### 2.5. GABA-Gated Chloride Channels

GABA-gated chloride channels are potential targets for insecticides. Once they are blocked by its antagonists (such as the α-Thujone (isolated from *Artemesia absinthium* L. (Asteraceae)) or the picrotoxine (*Anamirta cocculus* (L.) Wight and Arn (Menispermaceae)), neuronal inhibition is reduced, leading to hyper-excitation of the central nervous system (CNS), convulsion and death. Previous research have reported the monoterpene thujone as a neurotoxin, since it acts on GABAA receptors as an allosteric reversible modulator, and as a competitive inhibitor of [3H]Ethynylbicycloorthobenzoate ([3H]EBOB binding) [38]. Additionally, the GABA receptor may be inhibited by the monoterpenoids carvacrol, pulegone and thymol through [3H]TBOB binding [39]. Similarly, the silphinene-type sesquiterpenes, plant-derived natural compounds, antagonize the action of aminobutyricacid (GABA), by stabilizing non-conducting conformations of the chloride channel [24,38]. As GABA is an endogenous ligand related to stimulate feeding and evoke taste cell responses on most herbivorous insects, the chemicals that antagonize GABA receptors may also be considered as antifeedant or deterrent compounds, affecting mostly aphids, lepidopterans and beetles [40,41].

### 2.6. Octopamine Receptors

Octopamine is a multi-functional endogenous amine that acts as a neurotransmitter, neurohormone and neuromodulator on invertebrates [42]. Its receptors are widely distributed in the central and peripheral nervous systems of insects, comprising the octopaminergic system, constituting of several subtypes of octopamine receptors, which are coupled to different second messenger systems, therefore playing a key role in mediating physiological functions and behavioral aspects [43,44,45]. For instance, octopamine1 receptor modulates myogenic rhythm of contraction in locust extensor-tibiae through changes in intracellular calcium concentrations, whereas octopamine2A and octopamine2B receptors mediate their effects through the activation of adenylatecyclase. Moreover, octopamine3 receptors mediate changes in cyclic adenosine monophosphate (CAMP) levels in the locust central nervous system [46].

The rapid action of monoterpenes against some pests suggests a neurotoxic mode of action. This hypothesis was confirmed by Reynoso, et al. [47], who have demonstrated repellent and insecticidal activity of eugenol against the blood-sucking bug *Triatoma infestans* (Klug; Reduviidae) through activation of the octopamine receptor.

Previous studies have reported the presence of octopamine receptors in a large variety of insects, including, firefly, flies, nymphs, cockroaches and lepidopterans [46,47,48]. As these receptors do not conform to the receptor categories that have been recognized in vertebrates, agonists of octopamine receptors may be a valuable candidate for a commercial pesticide, once they are target-specific, less toxic to mammals and have a different mechanism of action when compared to the majority of pesticides currently in the market [47].

## 3. Plant Derived Insecticides That Affect Respiratory or Energy System

Cellular respiration is a process that converts nutrient compounds into energy or adenosine triphosphate (ATP) at a molecular level. More specifically, this process is performed by the electron transport chain of the mitochondria, which comprises several important enzymes that are potential targets for insecticides. Rotenone is the most common natural product among rotenoids, a type of isoflavonoid and is usually found in species from Derris and Lonchocarpus (in Fabaceae) and Rhododendron (in Ericaceae), spread throughout East Indies, Malaya and South America [20].

Rotenone is defined as a complex I inhibitor of the mitochondrial respiratory chain, which works both as contact and stomach poison. It blocks the nicotinamide adenine dinucleotide (NADH) dehydrogenase, stopping the flow of electrons from NADH to coenzyme Q, therefore, preventing ATP formation from NADH, but maintaining ATP formation through flavine adenine dinucleotide (FADH_2_); therefore, it is one of the slowest acting botanical insecticide, and yet readily degradable by air and sunlight, taking several days to kill insects, affecting primarily nerve and muscle cells, leading to cessation of feeding, followed by death, from several hours to a few days after exposure. Moreover, this bio-based pesticide is constantly applied to protect lettuce and tomato crops as it has a broad spectrum of activity against mite pests, including leaf-feeding beetles, lice, caterpillars, mosquitoes, ticks, fire ants and fleas. Furthermore, its effects are substantially synergized by PBO or pyrodone (MGK 264).

Rotenone is highly toxic to mammals and fish [24,49]. Its activity and persistence are comparable to dichlorodiphenyltrichloroethane (DDT) [2]; moreover, previous studies have correlated a possible link between its exposure and Parkinson’s Disease (PD) [50]. However, in spite of its high toxicity, rotenoids may be a potential source of novel complex I inhibitors, acting as a prototype for the development of safer and more efficient pesticide derivatives [51].

Acetogenins (annonins, asimicin, squamocin, annonacins) obtained from *Annona squamosa* L. (Annonaceae) are well known for their pest control properties. A botanical formulation based on annonins wherein asimicin is the major pesticidal compound has been patented [52].

## 4. Plant Derived Insecticides That Affect the Endocrine System

Chemical constituents that interfere with the endocrine system of insects are classified as insect growth regulators (IGR). They may act either as juvenile insect hormone mimics or inhibitors, as well as chitin synthesis inhibitors (CSI). Normally, the juvenile hormones are produced by insects in order to keep its immature state. When a sufficient growth has been reached, the production of the hormone stops, triggering the molt to the adult stage [53]. Triterpenes from *Catharanthus roseus* (L.) G. Don (Apocynaceae), such as α-amyrin acetate and oleanolic acid, have demonstrated interesting growth regulator activity [54]. Acyclic sesquiterpenes such as davanone, ipomearone and the juvenile hormone from silkworm are perfect examples of natural products with IGR activity as well. Therefore, the constant application of IGR towards the crops will maintain the insects in its larvae state, preventing a successful molting and resulting in an efficient pest control [55]. On the other hand, it has been reported the antijuvenile hormone activity of two chromenes found in *Ageratum conyzoides* L. (Asteraceae), precocene I and II promotes a precocious metarmophosis of the larvae and production of sterile, moribund and dwarfish adults after exposure [56]. Although, resistance to azadirachtin has been demonstrated [57], indicating that insects can develop resistance to natural hormones or hormone-related compounds; however, this class of compounds remains a natural potential for commercial bio-based pesticides [55]. Additionally, complex polyphenolic fractions also present a wide range of insecticidal activities, interfering with the fecundity and inducing the disruption of the oogenesis [58,59] (WO 94/13141).

Moreover, previous researches have reported a natural insecticide of broad-spectrum activity, which has low mammalian toxicity and is the least toxic among botanical insecticides. It is called azadirachtin, a complex tetranortriterpenoid limonoid, majorly found in the seeds of *Azadirachta indica* A. Juss. (Meliaceae), a plant species commonly known as the Neem tree which originated from Burma, but is currently grown in more arid, tropical and subtropical zones of Southeast Asia, Africa, Americas and Australia [24,26,60]. Azadirachtin is considered a contact poison of systemic activity, which may be categorized in two ways: direct effects towards cells and tissues, or indirect effects, represented by endocrine system interference. It is a powerful compound that acts mainly as a feeding deterrent and insect growth regulator, comprising a wide variety of insect taxa including Lepidoptera, Diptera, Hemiptera, Orthoptera, Hymenoptera [60]. As for its growth regulatory effects, azadirachtin affects the neurosecretory system of the brain insect, blocking the release of morphogenetic peptide hormones (e.g., prothoracicotropic hormone (PTTH) and allatostatins). These hormones control the function of the prothoracic glands and the corpora allata, respectively. Therefore, as the moulting hormone (which controls new cuticle formation and ecdyses) and the juvenile hormone (JH) (which controls the juvenile stage at each moult) are regulated by prothoracic glands and the corpora allata, any disruption on this biochemical cascade may lead to moult disruption, moulting defects or sterility. The effects on feeding, developmental and reproductive disruption are caused by effects of the molecule directly on somatic and reproductive tissues and indirectly through the disruption of endocrine processes [60].

Neem-based non-commercial products are normally found as neem oil, obtained from the cold pressing of its seeds, in order to control phytopathogens (including insects). The other product is a medium-polarity extract containing azadirachtin (0.2–0.6% of seed/weight) [2], whereas the actual commercial product is a 1 to 4.5% azadirachtin solution [61]. Despite its 20 h half-life, it ensures a reasonable persistence in field applications due to its systemic action [2].

In relation to CSIs, they inhibit the production of chitin, a β-(1,4)-linked homopolymer of N-acetyl-D-glucosamine, one of the most important structural components of nearly all fungi cell walls, and also a major component of the insect exoskeleton, which provides physical protection and osmoregulation. As chitin is absent on plant and mammalian species, while it is abundant in arthropods and most fungi, chitin biosynthesis has become an important target for developing more specific insecticides and antifungal agents. Previous research has reported chitin synthase inhibition activity of 2-benzoyloxycinnamaldehyde (2-BCA), a natural product isolated from the roots of *Pleuropterus ciliinervis* Nakai (Polygonaceae), which is a plant species traditionally used in Chinese folk medicine to treat inflammation and several types of infection [62].

## 5. Plant Derived Insecticides That Affect the Water Balance

Insects have a thin layer of wax covering their body, which provides the ecological function of preventing water loss from the cuticular surface. For instance, vegetable crude oils of rice bran, cotton seed and palm kernel, as well as saponins (natural soaps) may act by disrupting this protective waxy covering, affecting the water balance of insects through a rapid water loss from the cuticle, therefore leading to death by desiccation. Interestingly, the action of soaps affects the wax covering of insects [63]. The action of soaps on the wax covering of insects is influenced by the temperature [64]. Additionally, the crude oils may also act by interfering with insect respiration by plugging the orifices called spiracles, resulting in death by asphyxiation, controlling several types of insects such as whiteflies, mites, caterpillars, leafhoppers and beetles [1].

## 6. Other Classes of Pesticides

The botanical pesticide agents may also be categorized into repellents, attractants, antifeedants or deterrents, molluscisides, fungicides, phytotoxins (herbicides) and phototoxins [15]. These classes are less common in plant sources than the insecticides [65]. Sometimes, a given compound may act as an insecticide and/or as a repellent. The major difference between those two is that the repellent does not kill insects, but only keeps them away by releasing pungent vapors or exhibiting a slight toxic effect [66].

## 7. Repellents

There are several essential oils which are majorly constituted of monoterpenes and are considered extremely effective repellents, including lemongrass (*Cymbopogon flexuosus* (Nees) Will. Watson (Poaceae)), eucalyptus (*Eucalyptus globulus* Labill. (Myrtaceae)), rosemary (*Rosmarinus officinalis* L. (Lamiaceae)), vetiver (*Vetiveria zizanioides* (L.) Nash (Poaceae)), clove (*Eugenia caryophyllus* (Spreng.) Bullock and S.G. Harrison (Myrtaceae)) and thyme (*Thymus vulgaris* L. (Lamiaceae)) [67]. Catnip oil, for example, extracted from *Nepeta cataria* L. (Lamiaceae), is considered a highly effective repellent of mosquitoes, bees and other flying insects. As a matter of fact, this oil repels *Aedes aegypti* L. (Culicidae) ten times more than DEET, which is probably related to its most effective constituent, nepetalactone, a monoterpene lactone [68], which is also reported as a repellent for lady beetles, cockroaches, flies and termites [69,70]. The anthraquinone tectoquinone was also described as a repellent against termites [71,72], and alstonine alkaloid has a repellent and larvicidal activity against *Anopheles gambiae* Giles (Culicidae) [73].

## 8. Attractants

In relation to attractants, they are considered semio-chemicals or communication compounds, released by plants in order to attract insects or to attract natural predators of the insects that feed on the plant [74]. Miller [75] have related the release of (−) and (+) limonene from white pine (*Pinus strobus* L. (Pinaceae)) to the attraction of the white pine cone beetle, *Conophthorus coniperda* Schwarz (Curculionidae), as well as the attraction of the predator beetle, *Enoclerus nigripes* Say (Cleridae), through the release of (–)-α-pinene, as well as the sesquiterpene caryophyllene [76].

## 9. Antifeedants or Deterrents

Previous studies have correlated antifeedant activity to a chemoreception mechanism, consisting in the blockage of receptors that normally respond to phagostimulants or through stimulation of deterrent cells (primary antifeedancy). According to Qiao et al., [77] azadirachtin reduces the cholinergic transmission of neurons related to the suboesophageal ganglion (SOG) of *Drosophyla melanogaster* Meigen (Drosophylinae), which are strongly related to feeding behavior. Additionally, food consumption may also be reduced due to its toxic effects after the first intake (secondary antifeedancy), promoting astringency, bitter taste or anti-digestive activity to certain herbivores [78,79]. For instance, Okwute and Nduji [80] have reported that schimperii, a gallotannin isolated from *Anogeissus schimperi* (Hochst. ex Hutch and Dalziel) (Combretaceae) was responsible for conferring this unattractive taste to herbivores. Similar effects were reported to, isoflavonoids [81], acetogenines [82,83] or cyanogenic glycosides, such as linamarin [84].

Moreover, Lajide, Escoubas and Mizutani [66] have reported feed deterrent activity of ent-kaurane diterpenoids isolated from *Xylopia aethiopica* (Dunal) A. Rich. (Annonaceae), among which, (−)-kau-16-en-19-oic acid has demonstrated the strongest antifeedant activity. According to Okwute [2], 15-epi-4E-jatrogrossidentadione, a diterpene from *Jatropha podagrica* L. (Euphorbiaceae) have also demonstrated its antifeedant activity towards *Chilo partellus* Swinhoe (Crambidae). Moreover, silphinene sesquiterpenes (*Senecio palmensis* C. Sm. (Asteraceae)) and thymol (*Thymus vulgaris* L. (Lamiaceae)) have been described as model of insect antifeedants [40].

However, as demonstrated by Huang et al., in spite of numerous natural plant natural products acting as antifeedants, no commercial product based on this mode of action have been produced. Insect habituation to feeding deterrents considerably limits their utility in crop protection [85].

## 10. Phytotoxines or Herbicides

Regarding phytotoxins, they may be defined as natural herbicides that are naturally released by plant species in order to interfere with the growth or germination of specific targets around them, such as weeds, leaving the emitting plant with more chances to survive. In nature, such action is called allelopathy, and the compounds that promote this action are defined as allelopathic agents [86,87,88]. Clay, et al. [89] have reported a study regarding herbicidal activity of citronella oil against different weed species: the oil at a dose of 504 kg a.i. ha-1 largely killed the foliage of the weed species within one application. However, most species have regrown substantially after two months, except for *Senecio jacobaea* L. (Asteraceae), which was the most susceptible one. According to Ismail, et al. [76], its herbicidal activity occurs through inhibition of photosynthesis. Besides essential oils herbicidal activity, Ismail, Hamrouni, Hanana and Jamoussi [90] have also reported plant-derived isolated compounds, such as eugenol and 1,8-cineole, with herbicidal activity promoted through inhibition of DNA synthesis and mitosis. Furthermore, several classes of secondary metabolites have been already described as phytotoxins, including naphtoquinones, such as juglone [91,92], amino acids such as m-tyrosin e [93] and L-tryptophane [94], terpenoids as 5,6-dihydroxycadinan-3-ene-2,7-dione [2,95] and citronnellol [90], catechins [2,96], polyphenols [97] and alkylamides [98].

## 11. Phototoxins

There is a class of phytochemicals called phototoxins or light-activated compounds that instead of losing their efficiency due to sunlight degradation, they are actually increased or activated by two different mechanisms. In the first mechanism (less common), molecular oxygen from the phototoxin absorbs the energy from the light, generating activated species of oxygen which ultimately damage important biomolecules [99]. The other mechanism of action is photogenotoxic, where phytochemicals cause damage to DNA, triggered by sunlight activation, regardless of the presence of oxygen in the phototoxin. In actuality, thephototoxin on its ground state, absorbs the photon, reaching its excited state, which interacts with ground state O_2_ located in the tissue of its target, generating singlet oxygen and enabling insecticidal activity. This peculiar mode of action of photoxins is so different from conventional synthetic pesticides that cross resistence among them is unlikely [100,101].

Light-activated phototoxins may be exemplified by several classes such as quinones, furanocoumarins, substituted acetylenes and thiophenes. For instance, Marchant and Cooper [102] have reported several phototoxins, such as 3-methyl-3-phenyl-1,4-pentadiyne, an oil constituent from *Artemisia monosperma* Delile (Asteraceae), which under sunlight-induced conditions exerts an activity similar to DDT against the housefly *Musca domestica* L. (Muscidae) and cotton leaf worm *Spodoptera littoralis* Boisduval (Noctuidae) larvae. They have also discovered that an acetylenic epoxide from *Artemisia pontica* L. (Asteraceae), called ponticaepoxide, exhibits an LC_50_ of 1.47 ppm against mosquito larvae when submitted to UV light. Additionally, Nivsarkar, et al. [103] have also found that the major compound from the roots of *Tagetes minuta* L. (Asteraceae), a thiophene called terthiophene or α-terthienyl (αT), is highly toxic against several organisms when co-submitted to near UV light radiation, such as nematodes, red flower beetles (*Tribolium casteneum* Herbst (Neoptera)), blood-feeding insects such as *Manduca sexta* L. (Sphingidae) and mosquito larvae (dipteres): *Aedes aegypti* L. (Culicidae), *Aedes atropalpus* Coquillett (Culicidae), *Aedes intrudens* Dyar (Culicidae) and yet, to our knowledge no commercial product has been generated. Plant-based natural product chemical structures with their corresponding pesticide activity and targets may be found in Table 2.

## 12. Discussion

Since ancient times, efforts to protect the agricultural harvest against pests have been reported. The use of inorganic compounds to control pests was reported between 500 B.C and the 19th century. They included products based on sulphur, lead, arsenic and mercury [2]. On the other hand, plant biodiversity has proved to be an endless source of biologically active ingredients, used for traditional crop and storage protection. Egyptian and Indian farmers used to mix the stored grain with fire ash [104]. The ancient Romans used false hellebore (*Veratrum viride* Aiton (Melanthiaceae)) as a rodenticide. Moreover, pyrethrum (extract from *Tanacetum cinerariifolium* (Trevir.) Sch.Bip (Asteraceae)) was used as an insecticide in Persia and Dalmatia, whereas the Chinese have discovered the insecticidal properties of *Derris* spp. (Fabaceae) [105].

Previous studies have already reported more than 2500 plant species belonging to 235 families, which have demonstrated their biological activity against several types of pests [1]. However, in spite of the remarkable potential as natural sources for commercial botanical pesticide development, not many have been found on the market, remaining in use only for small organic crops and commonly classified as so-called farming products [106].

Plant-derived pesticides can be processed in various ways: as crude plant material in the form of dust or powder; as extracts or as pure plant natural products, formulated into solutions or suspensions [2]. Several different classes of natural compounds that promote pesticide activity have already been reported, namely: fatty acids, glycolipids, aromatic phenols, aldehydes, ketones, alcohols, terpenoids, flavonoids, alkaloids, limonoids, naphtoquinones, saccharides, polyolesthers, saponins and sapogenins [20,107,108,109,110]. However, several factors appear to limit the commercialization of botanical pesticides, such as: problems in large scale production, non-availability of raw materials, poor shelf life, diminished residual toxicity under field conditions, limitations regarding standardization and refinement of the final product. Additionally, as the phytochemical profile of plant species may vary according to its genome/transcriptome/proteome/metabolome, and this variation depends on several edaphic-climatic factors (i.e., temperature, relative humidity, level of sunlight radiation, altitude, photoperiod and type of soil) as well as ecologic interactions, (i.e., herbivory or mutualism), manufacturers must take additional care in order to maintain efficiency and ensure that their products will perform consistently (standardization). Finally, even if all these issues are addressed, regulatory approval remains as the major barrier. A serious drawback to commercialization of botanicals is the high cost of processing plant materials to meet the standards of pesticide regulatory authorities [111]. Marrone [112] provided an overview of the current state of biopesticides and offered some ideas for improving their adoption, including conducting on-farm demonstrations and more education and training on how the products work and how to incorporate them into integrated pest management. In many jurisdictions, no distinction is made between synthetic pesticides and biopesticides. Simply because a compound is a natural product, it does not mean that it is safe, since most of the toxic poisons are natural products or inspired by them. Furthermore, if biopesticides are used indiscriminately, as wells as the synthetics, they may also lead to the development of pest resistance [113]. 

In this context, only a few new sources of botanicals have reached commercial status in the past twenty years. Thus, the major commercial botanic pesticides currently in use include: pyrethrin, rotenone, azadirachtin and essential oils in general [111], along with three other products, commercialized in a more limited way: ryania, nicotine and sabadilla or veratrine alkaloids [20].

Therefore, the best strategy for a botanical pesticide to meet all the standards required and reach commercial status in a more efficient and pragmatic way is by performing bioassay-guided fractionation in a high scale [82,109,114]. In other words, the bioassays assessment of several plant extracts and its fractions, obtained whether by sophisticated or unsophisticated purification procedures, may lead to the discovery of the most effective compound or mixture of compounds correlated to the pesticide activity of each corresponding species. The isolated compound may act as a lead compound or prototype for the synthesis or semi synthesis of pesticide derivatives, which, by structure–activity relationship (SAR) techniques may result in more effective and safer products. However, sometimes, when the compound is presented on its isolated form, it may promote no activity at all, proving that extracts or fractions from a certain plant species are more effective than its isolated compounds, due to synergic effect of compound mixture, which may also lead to the manufacture of potential raw material for commercial biopesticides [114,115].

**Table 1 molecules-26-04835-t001:** Toxicity and mechanism of action of bio-based natural insecticides.

Product Name	Toxicity	Mammalian Toxicity (LD_50_ (mg/kg bw))	Reference	Mechanism of Action	Reference
Azadirachtin	IGR, R	13000 (Orally)	[116]	Prothoracicotropic hormone (PTTH) inhibitor; phagostimulant disruptor by cholinergic transmission reduction	[60]
Nicotine	C	50 (Orally)	[116]	Acetylcholine mimic; agonist of nicotinic acetylcholine receptor	[35]
Rotenone	S	350 (Orally)	[116]	Complex I inhibitor of the mitochondrial respiratory chain	[24,49]
Pyrethrins I and II	C, S	1200 (Orally)	[116]	Voltage-gated Sodium channels modulator	[21]
Ryania	C, S	750 (Orally)	[116]	Activation of sarcoplasmic reticulum calcium channels (ryanodine channels)	[11,29]
Sabadilla	C, S	5000 (Orally)	[116]	Voltage-gated Sodium channels modulator	[27]
*trans*-Cinnamaldeyde	C	1160 (Orally)	[116]	Inhibitor of β-(1,3)-glucan synthase and chitin synthase	[117]
1,8 cineole	C, F	2480 (Orally)	[116]	AChE inhibitor	[31]
Eugenol	C, F	500 (Orally)	[116]	Agonist of octopamine receptor	[47]
Citronella oil	C, F	7200 (Orally)	[89]	Inhibition of (AChE) and glutathione-S-transferase	[118]
Thujone	C, F	230 (Orally)	[119]	Allosteric reversible modulator of GABA_A_ receptors	[24,38,120]
Terthiophene	C	110 (intraperitoneally)	[121]	Light activated phototoxin. Activated species Oxygen generator	[102]
Palm kernel crude oil	C	>5000 (orally)	[122]	Disturbance of water balance caused by disruption of the protective waxy covering of insects	[1]

C—Contact. S—Stomach. F—Fumigant. IGR—Insect Growth Regulator. R—Repellent.

**Table 2 molecules-26-04835-t002:** Plant-based natural products with pesticide activity.

Compound/Trade Name	Plant Species	Class	Mode of Action	Targets	References
**Nervous system: Voltage-gated sodium channels**
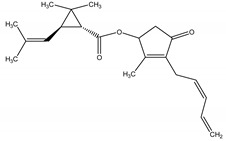 Pyrethrin I	*Tanacetum cinerariifolium* (Trevir.) Sch. Bip. (Asteraceae)	Cyclopropylmonoterpene esters	Insecticidal	Spider mites; flies; fleas; beetles	[23,24]
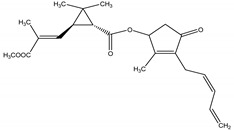 Pyrethrin II	*T. cinerariifolium*	Cyclopropylmonoterpene esters	Insecticidal	Spider mites; flies; fleas; beetles	[23,24]
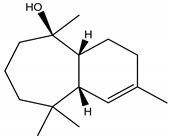 Himachalol	*Cedrus deodara* (Roxb. ex D. Don) G. Don (Pinaceae)	Sesquiterpene	Insecticidal	Pulse beetle (*Callosobruchus* *analis* Fabriciuss) and Housefly (*Musca domestica* L.)	[123]
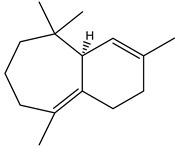 β-Himachalene	*C. deodara*	Sesquiterpene	Insecticidal	Pulse beetle (*Callosobruchus* *analis.*) and Housefly (*Musca domestica*)	[123]
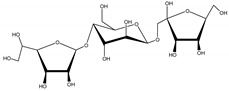 Decaleside II	*Decalepis hamiltonii* Wight and Arn. (Apocynaceae)	Trisaccharide	Insecticidal	Houseflies; cockroaches; stored grain pests	[25]
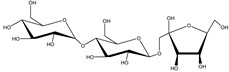 Decaleside I	*D. hamiltonii*	Trisaccharide	Insecticidal	Houseflies; cockroaches; stored grain pests	[25]
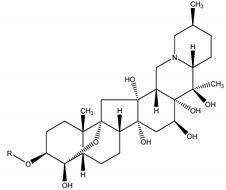 Cevadine R = (z)-CH_3_CH=C(CH_3_)COVeratridine R = 3,4-(CH_3_O)_2_PhCO	*Schoenocaulon officinale* (Schltdl. and Cham.) A. Gray ex Benth. (Melanthiaceae)	Sabadilla alkaloid or *Veratrum* alkaloid	Insecticidal	Stinks, leafhoppers, caterpillars; leafhoppers; housefly (*Musca domestica*) and thrips (*Scirtothrips* spp.)	[11,26,27,124]
**Nervous system: Voltage-gated calcium channels**
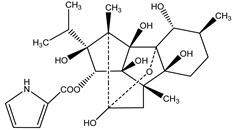 Ryanodine	*Ryania speciosa* Vahl (Salicaceae)	Diterpene	Insecticidal	Caterpillars; worms; potato beetle; lace bugs; aphids and squash bugs	[11,29]
**Acetylcholinesterase**
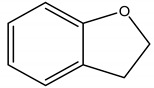 (Coumaran)2,3-dihydrobenzofuran	*Lantana camara* L. (Verbenaceae)	Benzofuran	Insecticidal	Stored grain pests (*Sitophilus* *oryzae* L.; *Tribolium castaneum herbst*); Housefly pests (*Musca domestica*)	[30]
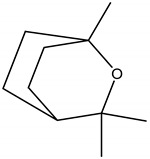 1,8-Cineole (Eucalyptol)	*Eucalyptus globulus* Labill. (Myrtaceae)	Monoterpene	Insecticidal	Head lice (*Pediculus humanus capitis* De geer)	[31]
**Nicotinic acetylcholine receptors**
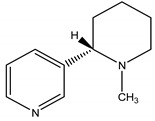 Nicotine	*Nicotiana tabacum* Velloso (Solanaceae)	Pyridine Alkaloid	Insecticidal Antifungal	Aphids; thrips; mites; leaf hoppers; spider mites; fungus	[24]
**GABA-gated chloride channels**
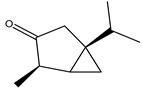 Thujone	*Artemisia absinthium* L. (Asteraceae); *Juniperus* sp. (Cupressaceae); *Cedrus* sp. (Pinaceae)	Monoterpene	Larvicidal	Western corn rootworm larvae (*Diabrotica virgifera*)	[38]
Insecticidal	Fruit fly (*Drosophila melanogaster* Meigen)
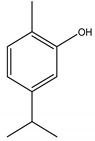 Carvacrol	**	Monoterpene	Insecticidal	*Periplaneta americana* L.	[39]
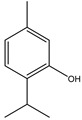 Thymol	**	Monoterpene	Insecticidal	*Periplaneta americana*	[39]
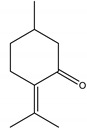 Pulegone	**	Monoterpene	Insecticidal	*Periplaneta americana*	[39]
**Octopamine receptors**
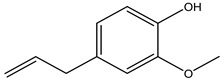 Eugenol	*Syzygium aromaticum* (L.) Merr. and L.M. Perry (Myrtaceae)	Monoterpene	Herbicidal	*Cassia occidentalis* and *Biden spilosa*	[90]
Insecticidal Repellent	Blood-sucking bug *Triatoma infestans* (Klug); fruit fly (*Drosophila melanogaster* (*Meigen*); American cockroach (*Periplaneta americana*)	[46,47,48]
**Respiratory or energy system**
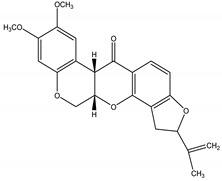 Rotenone	*Lonchocarpus* Kunth (Fabaceae); *Derris* Lour (Fabaceae); *Rhododendron* L. (Ericaceae)	Isoflavonoid/Rotenoid	Insecticidal; Piscicidal	Beetles; caterpillars; lice; mosquitoes; ticks; fleas; fire ants	[24,125]
**Endocrine system**
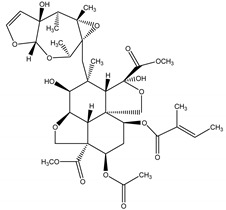 Azadirachtin	*Azadirachta indica* A. Juss. (Meliaceae)	Tetranortriterpenoid limonoid	Repellent; Antifeedant; Nematicidal; Sterilant; Antifungal; Insect Growth Regulator	Dandruffs eczema; stored grain pests; aphids; caterpillars; thrips; mealy bugs	[2,26,60,77,78,126]
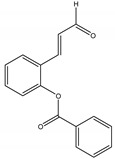 2’-Benzoyloxycinnamaldehyde (2’-BCA)	*Pleuropterus ciliinervis* Nakai (Polygonaceae)	Aromatic aldehyde	Antifungal	*Saccharomyces cerevisiae* Meyen ex Hansen.	[62]
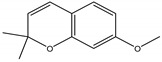 Precocene I (7-methoxy-2,2-dimethylchromene)	*Ageratum conyzoides* L. (Asteraceae)	Chromene	IGR	Sawtoothed grain beetle (*Oryzaephilus surinamensis* L.); Milkweed bug (*Oncopeltus fasciatu* Dallas); Noctuid moth (*Spodoptera litura* Fabricius); Parasitic wasp (*Microplitis rufiventris* Nees)	[127,128,129,130]
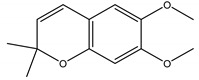 Precocene II (6,7-dimethoxy-2,2-dimethylchromene)	*A. conyzoides*	Chromene	IGR	Desert locust (*Schistocerca gregaria* Forskål); Milkweed bug (*Oncopeltus fasciatus Dallas*); Noctuid moth (*Spodoptera litura* Fabricius); Parasitic wasp (*Microplitis rufiventris* Nees)	[128,129,130,131]
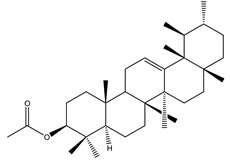 α-amyrin acetate	*Catharanthus roseus* (L.) G. Don (Apocynaceae)	Steroid	IGR	*Helicoverpa armigera* Hübner	[54]
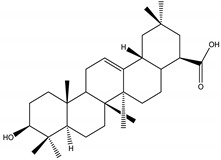 Oleanolic acid	*Catharanthus roseus* (L.) G.Don (Apocynaceae)	Steroid	IGR	*Helicoverpa armigera*	[54]
**Antifeedants**
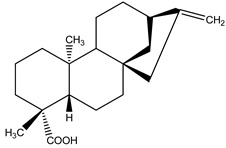 (−)Kau-16-en-19-oic acid	*Xylopia aethiopica* (Dunal) A. Rich. (Annonaceae)	Kaurane diterpene	Antifeedant	Termites (*Reticulitermes speratus* Kolbe)	[79]
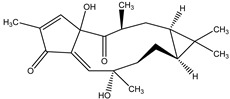 15-epi-4E-jatrogrossidentadione	*Jatropha podagrica* Hook. (Euphorbiaceae)	Diterpene	Antifeedant	Moth (*Chilo partellu* Swinhoes)	[2]
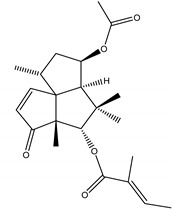 11-Acetoxy-5-isobutyryloxysilphinen-3-one	*Senecio palmensis* C. Sm. (Asteraceae)	Silphinene sesquiterpene	Antifeedant	Colorado potatobeetle (*Leptinotarsa decemlineata Say*);Aphids (*Myzus persicae* Sulzer, *Diuraphis noxia* Kurdjumov, *Rhopalosiphum padi L.*, *Metopolophium dirhodum Walker*, *Sitobiona venae* Fabricius)	[40]
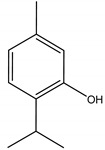 Thymol	*Thymus vulgaris* L. (Lamiaceae)	Monoterpene	Antifeedant	Colorado potato beetle (*Leptinotarsa decemlineata*);Aphids (*Myzus persicae*, *Diuraphis noxia*, *Rhopalosiphum padi*, *Metopolophium dirhodum*, *Sitobiona venae*)	[40]
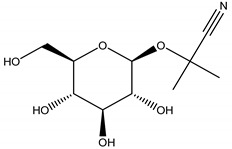 Linamarin	*Lotus corniculatus* L. (Fabaceae); *Trifolium repens* L. (Fabaceae)	Cyanogenic glycoside	Antifeedant	Snails (*Arianta arbustorum* L. and *Helix* aspersa O.F. Müller; slugs (*Agriolimax reticulates* O.F. Müller); lemmings (*Lemmus lemmus* L.); aphids (*Aphis craecivora* Koch, *Nearctaphis bakeri* Cowen ex Gillette and Baker)	[84]
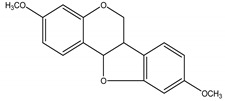 (–)-homopterocarpin	*Pterocarpus macrocarpus* Kurz (Fabaceae)	Isoflavonoid/Pterocarpan	Antifeedant	Common cutworm (*Spodoptera litura* F.) and the subterranean termite (*Reticulitermes speratus*)	[81]
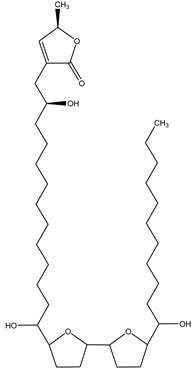 Asimicin	*Asimina triloba* (L.) Dunal (Annonaceae)	Polyketide/Acetogenin	Insecticidal; Antifeedant	Mexican bean beetle (*Epilachna varivestis* Mulsant); striped cucumber beetle (*Acalymma bivittatum Fabricius*); two-spotted spider mite (*Tetranychus urticae* Koch); melon aphid (*Aphis gossyphii Glover*)	[82,83]
Nematicidal	*Caenorhabditis elegans* (Maupas)
Larvicidal	Blowfly larvae (*Calliphora vicina* Robineau-Desvoidy); mosquito larvae (*Aedes aegypti*)
**Repellents**
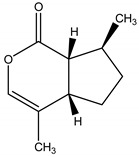 Nepetalactone	*Nepeta cataria* L. (Lamiaceae)	Monoterpene lactone	Repellent	Mosquitoes (*Aedes aegypti*); bees;lady beetle; cockroaches; flies; termites	[68,69,70]
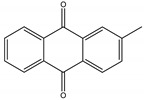 Tectoquinone	*Tectona grandis* L. f. (Lamiaceae)	Anthraquinone	Repellent	Termites (*Cryptotermes brevis* Walker and *Reticulitermes flavipes* Kollar)	[71,72,132]
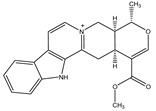 Alstonine	*Alstonia boonei De* Wild. (Apocynaceae)	Indoloquinolizidine alkaloid	Repellent; Larvicidal	Mosquito (*Anopheles gambiae Giles*)	[73]
**Attractants**
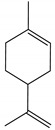 Limonene	*Pinus strobes* D. Don (Pinaceae)	Monoterpene	Attractant	White pine cone beetle (*Conophthorus coniperda* Schwarz)	[75]
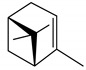 (–)-α–Pinene	*P. strobus*	Monoterpene	Attractant	*Enoclerus nigripes Say*	[75]
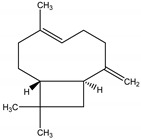 (*E*)-β-caryophyllene	*Zea mays* L. (Poaceae)	Sesquiterpene	Attractant	Nematodes (*Heterorhabditis megidis* Poinar, Jackson and Klein), natural enemy/parasite of corn root worm (*Diabrotica virgifera* Leconte)	[76]
**Phytotoxins**
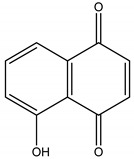 Juglone	*Juglans nigra* L. (Juglandaceae)	Naphtoquinone	Herbicidal	*Echinochloa crus-galli* L.; *Amaranthus retroflexus* L.; *Abutilon theopharasti* Medik	[91,92]
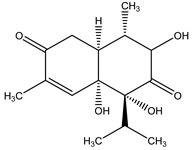 5,6-dihydroxycadinan-3-ene-2,7-dione	*Eupatorium adenophorum* Spreng. (Asteraceae)	Cadinane sesquiterpene	Herbicidal	*Arabidopsis thaliana* (L.) Heynh	[95]
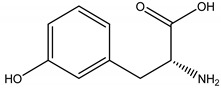 m-Tyrosine	Poaceae spp.	Amino acid	Herbicidal	Weeds	[93]
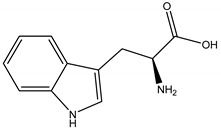 Tryptophan	*Prosopis juliflora* (Sw.) (Fabaceae)	Amino acid	Herbicidal	Barnyard grass (*Echinochloa crus-galli* L.)	[133]
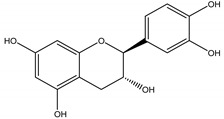 (−)-Catechin	*Centaurea stoebe* L. (Asteraceae)	Flavanol	Herbicidal	*Koeleria macrantha* (Ledeb.) Schult., and *Festuca idahoensis* Elmer	[96]
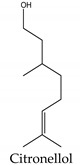 Citronellol	*Cymbopogon citratus* (DC.) Stapf (Poaceae)	Monoterpene	Herbicidal	*Cassia occidentalis* L.	[90]
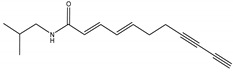 (*E*,*E*)-2,4-undecadien-8,10-diynoic acid isobutylamide	*Acmella oleracea* (L.) R. K. Jansen (Asteraceae)	Isobutylamide	Herbicidal	Cress (*Lepidum sativum* L.) and barnyard grass (*Echinochloa crus-galli* (L.) P. Beauv)	[98]
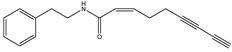 Nona-(2*Z*)-en-6,8-diynoic acid 2-phenylethylamide	*A. oleracea*	Phenylethylamide	Herbicidal	Cress (*Lepidum sativum*) and barnyard grass (*Echinochloa crus-galli*)	[98]
**Antifungals**
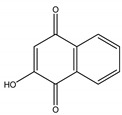 Lawsone	*Lawsonia inermis* L. (Lythraceae)	Naphtoquinone	Antifungal	*Alternaria solani* (Fr.) Keissl.; *Alternaria tenuis* Nees; *Aspergilus niger* Tieghem; *Aspergilus wentii* Whemer; *Absidia ramosa* (Zopf) Vuil; *Absidia corymbifera* Cohn; *Acrophialophora fusispora* (S.B. Saksena) Samson; *Circinella umbellate* Tiegh. and G. Le Monn	[134]
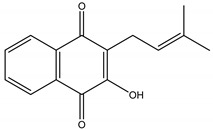 Lapachol	*Tabebuia serratifolia* (Vahl) G. Nicholson (Bignoniaceae)	Naphtoquinone	Antifungal; Larvicidal	*Aedes aegypti*; *Gloeophyllum trabeum* ATCC 11539; *Trametes versicolor* (L.) Lloyd	[135,136]
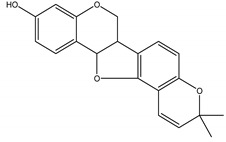 Phaseollin	*Z. mays*	Isoflavonoid/Pterocarpan	Antifungal	*Botrytis cinerea* Pers.; *Colletotrichum lindemuthianum* (Sacc. and Magnus) Briosi and Cavara,; *Fusarium solani* Mart.; *Rhizoctonia solani* J. G. Kühn and *Thielaviopsis basicola* (Berk. and Broome) Ferraris	[137,138]
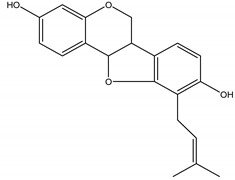 Phaseollidin	*Z. mays*	Isoflavonoid/Pterocarpan	Antifungal	*Botrytis cinerea*; *Colletotrichum lindemuthianum*; *Fusarium solani*; *Rhizoctonia solani* and *Thielaviopsis basicola*	[137,138]
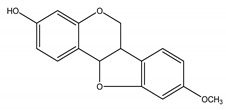 Medicarpin	*Medicago sativa* L. (Fabaceae)	Isoflavonoid/Pterocarpan	Antifungal	*Colletotrichum phomoide* (Sacc.) Chester.; *Stemphylium loti* J.H. Graham,; *Stemphylium botryosum* Walroth; *Phoma herbarum* Westendorp; *Leptosphaeria briossiana* (Higgings) and *Cladosporium cladosporoides* (Fresen.) G.A. de Vries	[137,138]
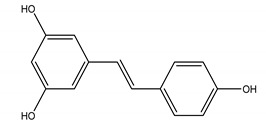 Resveratrol	*Polygonum cuspidatum* Siebold and Zucc. (Polygonaceae)	Polyphenol/Stilbene	Anti-viral	Tobacco Mosaic Virus (TMV)	[97]
Antifungal	*Alternaria solani*; *Botrytis cinerea*; *Fusarium graminearum* Schwabe; *Phytophthora capsici Leonian*; *Phytophthora infestans* (*Mont.*) *de Bary*; *Rhizoctonia* solani J.G. Kühn; *Sclerotinia sclerotiorum* (Lib.) (Y. Nisik. and C. Miyake) Shoemaker, *Rhizoctonia cerealis* D. I. Murray and Burpee; *Watermelon anthracnose*
Herbicidal	*Digitaria sanguinalis* (L.) Scop.; *Echinochloa crus-galli*
Insecticidal	Oriental armyworm (*Mythimna separata* Walker); Cotton bollworm (*Helicoverpa armigera*); Corn borer (*Ostrinia nubilalis* Hubner)
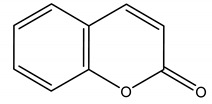 2H-chromen-2-one	*Lavandula angustifolia Mill.* (*Lamiaceae*)	Polyphenol/Coumarin	Antibacterial	*Ralstonia solanacearum* Smith	[139]
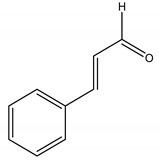 *Trans*-Cinnamaldehyde	*Cinnamomi cortex* J. Presl (Lauraceae)	Aromatic aldehyde	Antifungal	*Saccharomyces cerevisiae*	[117]
**Phototoxins**
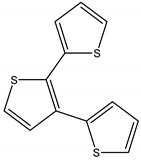 Terthiophene	*Tagetes minuta* L. (Asteraceae)	Thiophene	Nematicidal	Nematodes	[103]
Insecticidal	Tobacco hornworm (*Manduca sexta*); Lepidopteran (*Pieris rapae* L.); housefly (*Musca domestica*); Red flower beetle (*Tribolium casteneum Herbst*); mosquito larvae (*Aedes atropalpus*, *Aedes aegypti* and *Aedes intrudens*)
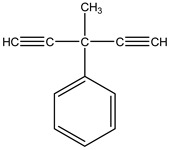 3-methyl-3-phenyl-1,4-pentadiyne	*Artemisia monosperma* Delile (Asteraceae)	Substituted Acetylene	Insecticidal	Housefly (*Musca domestica*) and Cotton Leaf worm (*Spodoptera littoralis* Boisduval)	[102]

** Purchased from Sigma–Aldrich Chemical Co.

## 13. Conclusions

Despite several factors that appear to limit botanical pesticide commercialization, such as problems in large scale production, non-availability of raw materials, poor shelf life, diminished residual toxicity under field conditions and lack of extract standardization, a multidisciplinary approach, comprising bioassay-guided fractionation, combined with structure-activity relationship (SAR) and analytical techniques, has revealed to be an extremely efficient strategy in order to develop bio-based pesticides that meet all the commercial standards required. In summary, plant-derived pesticides have indicated their potential as a great alternative for pest management, once they become cheap, target-specific, less hazardous to human health, biodegradable and ecofriendly; therefore, they may improve crop efficiency and reduce food crisis while maintaining sustainability.

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
