# Peer review of "Plant-Derived Pesticides as an Alternative to Pest Management and Sustainable Agricultural Production: Prospects, Applications and Challenges"

_molecules, 2021, doi:10.3390/molecules26164835_

Round 1

Reviewer 1 Report

Review of MS-1299282 -Molecules “Plant-Derived Pesticides as an Alternative to Pest Management and Sustainable Agricultural Production: Prospects, Applications and Challenges”.

The authors have made a considerable effort to present a varied group of derivatives from plants used as pesticides, focusing attention on sustainable production in agriculture. The approach adopted is interesting as the authors also try to explain the modes of action of the different substances. Authors must pay particular attention to the bibliography, as it is a review article

The authors can introduce along their MS which of these plant derivatives are allowed in the biological control regulations. Obviously, comparisons to the main countries (EU, China, US, Brazil, Turkey) can be limited to summaries. This information would be very useful for the reader.

Line 21-23. Rewrite the sentence in a more general sense as a list of substances presented in this way may not be exhaustive and include all substances.

Line 40-41. Be more precise in this estimate. Although the authors report the bibliographic reference, it is also advisable to report the period considered and the sources;

Line 54-56. The authors can insert a bibliographic reference for the negative effects of the molecule there is a robust scientific production especially in the medical and neurotoxicological fields;

Line 69-72. I would also add, in addition to the awareness achieved, that in the various countries and also in the EU the legislation has become increasingly stringent and binding with the consequences of a green choice in the use of pesticides;

Line109. The species name Triatoma infestans (Klug) in italics and add the family;

Page 5 line 6. Paragraph 3 should be a subparagraph of the 2? It is not clear!

Pag 5 Paragraph 3. Specify that this mode of action is very similar (modulators on voltage-gated sodium channels) also to other synthetic insecticides;

Pag 5 Paragraph 4. Specify if Ryania or its derivatives (they are also used in other countries eg India, EU, China);

Page 5-6. Line 47-52. Insert bibliographic reference for the use of coumaran extracted from Lantana (see doi: 10.1155 / 2014/187019);

Pag 6. Paragraph 7. Insert some examples of botanical species that contain GABA antagonist;

Pag 8. Paragraph 10. Specify better the physiological properties of azadirachtin, including effects on insect reproduction;

Page 8. Line 187. Do the authors also include insects with the term phytopathogens?

Pag 9. Line 236 – 237, 245, 258. The species name in italics and add the family for insect species;

Page 9. Paragraph 11. In this paragraph the authors must support with more bibliography. This paragraph is also interesting because the action of soaps on the wax covering of insects is influenced by the temperature;

Line 226. For Aedes aegypti enter the name of the family and order, as well as every time (eg line 237, 238, 299, 300, etc) that the names of the insect species are entered.

Line 263-263. Insert reference for example https://doi.org/10.1111/j.1570-7458.1995.tb01973.x

Line 299-300. Use italics for all species and family name;

Line 307. Use italics for all species and family name;

In the discussion, also introduce some aspects that are not mentioned in the MS, such as the biopesticidal effects also against useful insects such as predators (example doi: 10.2478 / V10045-012-0003-X).

Table 2 requires better formatting. In particular, the figures are sometimes superimposed with the name plant species.

In table 2  don’t use italics in “and Housefly”

Author Response

Dear Editor

           Please find enclosed the revised manuscript entitled  “Plant-Derived Pesticides as an Alternative to Pest Management and Sustainable Agricultural Production: Prospects, Applications and Challengesfor publication in Molecules, as a review paper.

            I would like to thank the referees for the positive evaluation, suggestions, comments and corrections. Their insights made relevant contributions to improve the quality of the manuscript. All suggestions and comments were addressed as detailed bellow.   

Sincerely yours,

Dr. Gerardo Cebrián-Torrejón

Review of MS-1299282 -Molecules “Plant-Derived Pesticides as an Alternative to Pest Management and Sustainable Agricultural Production: Prospects, Applications and Challenges”.

The authors have made a considerable effort to present a varied group of derivatives from plants used as pesticides, focusing attention on sustainable production in agriculture. The approach adopted is interesting as the authors also try to explain the modes of action of the different substances. Authors must pay particular attention to the bibliography, as it is a review article

Authors: We appreciate the positive comments.

The authors can introduce along their MS which of these plant derivatives are allowed in the biological control regulations. Obviously, comparisons to the main countries (EU, China, US, Brazil, Turkey) can be limited to summaries. This information would be very useful for the reader.

Authors: We agree and a new paragraph was introduced.

Line 21-23. Rewrite the sentence in a more general sense as a list of substances presented in this way may not be exhaustive and include all substances.

Authors: We agree and the sentence has been modified.

Line 40-41. Be more precise in this estimate. Although the authors report the bibliographic reference, it is also advisable to report the period considered and the sources;

Authors: We agree and the refereces were introduced.

Line 54-56. The authors can insert a bibliographic reference for the negative effects of the molecule there is a robust scientific production especially in the medical and neurotoxicological fields;

Authors: We agree and the refereces were introduced.

Line 69-72. I would also add, in addition to the awareness achieved, that in the various countries and also in the EU the legislation has become increasingly stringent and binding with the consequences of a green choice in the use of pesticides;

Authors: We agree and a new paragraph was introduced and reference were introduced.

Line109. The species name Triatoma infestans (Klug) in italics and add the family;

Authors: We agree and the MS was modified .

Page 5 line 6. Paragraph 3 should be a subparagraph of the 2? It is not clear!

Authors: We agree and a the MS was modified.

Pag 5 Paragraph 3. Specify that this mode of action is very similar (modulators on voltage-gated sodium channels) also to other synthetic insecticides;

Authors: We agree and a the MS was modified.

Pag 5 Paragraph 4. Specify if Ryania or its derivatives (they are also used in other countries eg India, EU, China);

Authors: We agree and a the MS was modified.

Page 5-6. Line 47-52. Insert bibliographic reference for the use of coumaran extracted from Lantana (see doi: 10.1155 / 2014/187019);

Authors: We agree and a the MS was modified and reference introduced

Pag 6. Paragraph 7. Insert some examples of botanical species that contain GABA antagonist;

Authors: We agree and a the MS was modified.

Pag 8. Paragraph 10. Specify better the physiological properties of azadirachtin, including effects on insect reproduction;

Authors: We agree and a the MS was modified.

Page 8. Line 187. Do the authors also include insects with the term phytopathogens?

Authors: We agree and a the MS was modified.

Pag 9. Line 236 – 237, 245, 258. The species name in italics and add the family for insect species;

Authors: We agree and a the MS was modified.

Page 9. Paragraph 11. In this paragraph the authors must support with more bibliography. This paragraph is also interesting because the action of soaps on the wax covering of insects is influenced by the temperature;

Authors: We agree and a the MS was modified and the references were introduced.

Line 226. For Aedes aegypti enter the name of the family and order, as well as every time (eg line 237, 238, 299, 300, etc) that the names of the insect species are entered.

Authors: We agree and a the MS was modified.

Line 263-263. Insert reference for example https://doi.org/10.1111/j.1570-7458.1995.tb01973.x

Authors: We agree and the reference was introduced

Line 299-300. Use italics for all species and family name;

Authors: We agree and a the MS was modified.

Line 307. Use italics for all species and family name;

Authors: We agree and a the MS was modified.

In the discussion, also introduce some aspects that are not mentioned in the MS, such as the biopesticidal effects also against useful insects such as predators (example doi: 10.2478 / V10045-012-0003-X).

Authors: We agree and a the MS was modified and more reference were introduced

Table 2 requires better formatting. In particular, the figures are sometimes superimposed with the name plant species.

Authors: We agree and a the table 2 was modified.

In table 2  don’t use italics in “and Housefly”

Authors: We agree and a the table 2 was modified.

Reviewer 2 Report

The manuscript " Plant-derived pesticides as an alternative to pest management and sustainable agricultural production: prospects, applications and challenges" by Gerardo et al., presents a review? For biopesticides and especially the plant products with insecticidal activity to insects. Authors made an interesting work and obtained valuable information biopesticides but unfortunately, in my opinion, the manuscript in its current form does not meet the standards of the Molecules journal. In this sense I think the text still requires for a strong revision before submission to this or any other journal. I still consider that authors made an important work. In this sense I encourage them to strongly rework the current version (they may ask for help from an entomologist or pesticide specialist) of the document before re-submitting to this or another journal. References are lacking by good well known key papers that must be included.

Below I give the authors more specific comments of the manuscript that hopefully help them to improve it.

L39,42 insects and pest? Pest include insects.

L44-45, Control pest and diseases not eliminations,

L46-68, The authors review only CLD or DDT. I would like to read about more recent class of insecticides, (especially neonicotinoids), the effect of those insecticides on secondary pest,  bees (imidacloprid ban from EU), biological control agents, and finally to mammals, fish and other plant species (the authors may read Skouras et al 2019 Chemosphere).

L74 target specific (more than chemical insecticides) because natural pyrethrin’s have the same mode of action as chemicals PY.

P5L8 Please use italics for each plant or other species. Please correct it in whole manuscript.

Author Response

 Please find enclosed the revised manuscript entitled  “Plant-Derived Pesticides as an Alternative to Pest Management and Sustainable Agricultural Production: Prospects, Applications and Challengesfor publication in Molecules, as a review paper.

            I would like to thank the referees for the positive evaluation, suggestions, comments and corrections. Their insights made relevant contributions to improve the quality of the manuscript. All suggestions and comments were addressed as detailed bellow.   

Sincerely yours,

Dr. Gerardo Cebrián-Torrejón

Review of MS-1299282 -Molecules “Plant-Derived Pesticides as an Alternative to Pest Management and Sustainable Agricultural Production: Prospects, Applications and Challenges”.

The manuscript " Plant-derived pesticides as an alternative to pest management and sustainable agricultural production: prospects, applications and challenges" by Gerardo et al., presents a review? For biopesticides and especially the plant products with insecticidal activity to insects. Authors made an interesting work and obtained valuable information biopesticides but unfortunately, in my opinion, the manuscript in its current form does not meet the standards of the Molecules journal. In this sense I think the text still requires for a strong revision before submission to this or any other journal. I still consider that authors made an important work. In this sense I encourage them to strongly rework the current version (they may ask for help from an entomologist or pesticide specialist) of the document before re-submitting to this or another journal. References are lacking by good well known key papers that must be included.

Authors: We appreciate the positive comments and a strong revision of the MS was done in order to satisfity the referee. A expert in insect research was consulted and the manuscript was reviewed.

Below I give the authors more specific comments of the manuscript that hopefully help them to improve it.

L39,42 insects and pest? Pest include insects.

Authors: We agree and a the MS was modified.

L44-45, Control pest and diseases not eliminations,

Authors: We agree and a the MS was modified.

L46-68, The authors review only CLD or DDT. I would like to read about more recent class of insecticides, (especially neonicotinoids), the effect of those insecticides on secondary pest,  bees (imidacloprid ban from EU), biological control agents, and finally to mammals, fish and other plant species (the authors may read Skouras et al 2019 Chemosphere).

Authors: We agree and a the MS was modified and references were introduced.

L74 target specific (more than chemical insecticides) because natural pyrethrin’s have the same mode of action as chemicals PY.

Authors: We agree and a the MS was modified.

P5L8 Please use italics for each plant or other species. Please correct it in whole manuscript.

Authors: We agree and a the MS was modified.

Round 2

Reviewer 1 Report

Dear authors,

You have accepted the notes and indications reported in the previous version of the MS. I reiterate that the work is very interesting and I encourage you to continue improving your research activities in this area.

Reviewer 2 Report

I have read the revised manuscript and author's response to reviewer comments. The authors have made adequate responses to each item of comments and also make reasonable revision. I have no more comments

Just change ref 12 Panagiotis JS to Skouras PJ